# A Surgical Pen-Type Probe Design for Real-Time Optical Diagnosis of Tumor Status Using 5-Aminolevulinic Acid

**DOI:** 10.3390/diagnostics11061014

**Published:** 2021-06-01

**Authors:** Kicheol Yoon, Kwanggi Kim, Seunghoon Lee

**Affiliations:** 1Department of Biomedical Engineering, College of Medicine, Gachon University, 38-13, Dokjom-ro 3, Namdong-gu, Incheon 21565, Korea; kcyoon98@gachon.ac.kr; 2Medical Devices R&D Center, Gachon University Gil Hospital, 21, 774 beon-gil, Namdong-daero Namdong-gu, Incheon 21565, Korea; 3Department of Biomedical Engineering, College of Health Science, Gachon University, 191 Hambakmoero, Yeonsu-gu, Incheon 21936, Korea; 4Department of Health Sciences and Technology, Gachon Advanced Institute for Health Sciences and Technology (GAIHST), Gachon University, 38-13, 3 Dokjom-ro, Namdong-gu, Incheon 21565, Korea; 5Department of Neurosurgery, Daejeon Eulji Medical Center (Eulji University Hospital), Dunsanseo-ro, Seo-gu, Daejeon 35233, Korea; nslsh@eulji.ac.kr; 6School of Medicine, Eulji University, 77 Gyeryong-ro 771 Beon-gil, Jung-gu, Daejeon 34824, Korea

**Keywords:** fluorescence microscopy, fluorescence emission, malignant tumor, diagnosis, animal experiment

## Abstract

A surgical microscope is large in size, which makes it impossible to be portable. The distance between the surgical microscope and the observation tissue is 15–30 cm, and the adjustment range of the right and left of the camera is a maximum of 30°. Therefore, the surgical microscope generates an attenuation (above 58%) of irradiation of the optical source owing to the long working distance (WD). Moreover, the observation of tissue is affected because of dazzling by ambient light as the optical source power is strong (55 to 160 mW/cm^2^). Further, observation blind spot phenomena will occur due to the limitations in adjusting the right and left of the camera. Therefore, it is difficult to clearly observe the tumor. To overcome these problems, several studies on the handheld surgical microscope have been reported. In this study, a compact pen-type probe with a portable surgical microscope is presented. The proposed surgical microscope comprises a small and portable pen-type probe that can adjust the WD between the probe and the observed tissue. In addition, it allows the adjustment of the viewing angle and fluorescence brightness. The proposed probe has no blind spots or optical density loss.

## 1. Introduction

The 5-aminolevulinic acid (5-ALA) fluorescence dye is used to assess the condition of tumor removal in gliomas, brain tumor, breast cancer, colonic mucosa, and ovarian cancer [1,2,3,4,5]. In particular, malignant tumors are complex and are composed of tumors and blood vessels. Tumors and vasculature have the same color, making it difficult to differentiate the boundary between them using the naked eye. Therefore, a fluorescence microscope for surgery is used.

Fluorescent microscopes for surgery can easily distinguish the boundary between tumors and blood vessels, the shape of tumors and blood vessels, and the color of tumors and blood vessels through fluorescent staining on tumors and blood vessels [6,7]. However, these guidance cameras have disadvantages as they are large and heavy (camera head: above 4 kg, entire: above 6 kg) with limited portability [8,9,10,11,12,13,14]. In addition, they only allow for a limited adjustment of the observed beam angles (maximum 30°) and the working distance (WD) of 15–30 cm/cm^2^ [8,9,10,11,12,13,14,15,16,17]. Because a surgical camera uses a high-energy light source (55–160 mW/cm^2^) for the electromotive force of fluorescence emission, optical source beam energy loss (above 58%) can occur at the WD (15–30 cm), and strong ambient light can be observed around the tissue (photobleaching) [8,16,17,18].

To overcome these problems, several literatures regarding the handheld surgical microscope have been reported [19,20,21,22,23,24,25,26,27].

The advantage of the handheld type of surgical microscope is that it is smaller than the conventional surgical microscope, and the WD is shorter than the conventional one. In addition, the handheld type of surgical microscope is free of the photographing angle. However, the size of the handheld type of surgical microscope remains problematic [19,25,26]. Although the microscope in the study in [25] has a light mass, its size is long. In addition, while the WD of the microscope in the study in [20] that uses the image-guided raman spectroscopy technique is short, the microscope in the study in [20] still requires an optical light source that is capable of producing high energy (>600 mW @ 405 nm) for excitation. Furthermore, it is inconvenient to use the microscope in the study in [20] in surgery because the system is not equipped with either an optical source or a camera; they are independent of each other. Moreover, the construction of a system is complex. Therefore, the system is inconvenient for the surgeon to use the camera and the light source, respectively. It also takes a lot of hands for the operation.

To address the abovementioned problems, this study aimed to develop a pen-type surgical fluorescence microscope that is small and portable with an adjustable beam angle and WD. It provides a real-time diagnosis. The long-pass filter and camera module are self-fabricated, and it was measured for verification of excellent performance. A proposed pen-type probe is used for the CMOS camera which has a small size, a simple structure, low power consumption, a wide FOV, and a fast response time. The performance of the proposed pen-type probe was verified through animal experiments by using a light source of 405 nm (excitation wavelength) and a fluorescence wavelength (emission) of 620 to 670 nm. The aim of the animal test was to obtain the possibility of Protoporphyrin IX (PpIX) emission by irradiation of 405 nm at the tumor. 

## 2. Design of the Pen-Type Probe

The conventional surgical microscope (with a laser diode (LD: Thorlabs DL-5146-101s) and near-infrared (NIR) camera: SE8J200) and the proposed pen-type probe are compared in Figure 1 [18]. Figure 2 shows the detailed structure of the proposed pen-type probe. As shown in Figure 2, it comprises a laser diode, a drive module (for laser drive), a small camera module, a switch (ON/OFF), a laser brightness control performance, a battery, a long-pass filter, and a communication cable. The communication cable connects the camera and the external monitor. Therefore, the videos captured by the camera can be read in real time on an external monitor through a wire.

The excitation wavelength of an LD is 405 nm, and the bias voltage/current, output power, and beam divergence (θ_div_) are 5.6 V/100 mA, 19.0 mW/cm^2^, and 16°, respectively. The laser brightness control performance was controlled using a variable resistor (max. 10 kΩ). Therefore, the LD can adjust the excitation power. The endoscope camera, whose module had the bias voltage of 3 V, was used in the CMOS sensor. In addition, the horizontal length and diameter of the camera dimensions were 4 mm and 1.5 mm, respectively, as shown in Figure 3a.

Figure 3 shows the resolution, imaging frame, pixel size, field of view (FOV), and sensor type of the camera were 1600 × 1200, 30 fps, 2.0 M_pixel_, 60°, and CMOS sensor type, respectively. The irradiation beam angle of an LD can be adjusted by 360°. In addition, the WD and beam focus angle (θ_f_) of the camera were 50 mm and 9.5°, respectively (see Figure 1). Then, the WD was adjusted from 0 to 50 cm, and the realization environment was measured to be 50 mm.

The camera lens (head) was connected to a long-pass filter with a cut-in wavelength of 600 nm, as shown in Figure 3a. Therefore, the long-pass filter was fabricated using titanium oxide (T_i3_O_5_), silicon oxide (S_i_O_2_), and a glass-coated substrate, as shown in Figure 3b. Figure 3 shows that the *t*, *l*, and *h* of the fabricated filter were 2, 4, and 0.4 μm, and the diameter of the filter was 1.5 mm.

Since the PpIX (5-ALA) emission wavelength range is 620–670 nm, it can pass through a long-pass filter which only transmits wavelengths above 600 nm. Therefore, the long-pass filter can be used in fluorescence observation of PpIX upon the irradiation of PpIX with a light source at 405 nm. Figure 4 shows the fabrication of a pen-type probe that was created via the 3D printing technique.

The overall size and mass of the pen-type probe were 17 mm × 2.5 mm and 32 g (±2 g), respectively; the mass of the conventional surgical microscope is 6.0 kg [13,16]. Thus, the pen-type probe is significantly lighter than the conventional surgical microscope.

## 3. Experimental Results

The measurement results for the fabrication of the long-pass filter are shown in Figure 5. Figure 5 shows that the cut-in wavelength, transmission (T), and reflection (OD: optical density) were 600 nm, 98%, and 0.2%, respectively. 

A small animal (rat) experiment was performed to test the performance of the produced pen-type probe. The animal was tested from the Experimental Animal Center of Korea’s Lee, Gil-Ya Cancer Diabetes Research Institute; the IACAU number is LCDI-2017-0050. We requested permission from the animal ethics commission of the animal institutional review board (IRB). The species, age, weight, and strains were rat (male), 8 weeks, 240 g, and Sprague Dawley (SD), respectively.

The 5-ALA substance generated the Protoporphyrin IX (PpIX) material after 5 to 6 h through the chemical reaction with the tumor when medication was taken (oral intake) as shown in Figure 6a,b [17]. At this time, the PpIX had no response to the fluorescence emission property. However, when the external light source was irradiated in the PpIX, the PpIX emitted the fluorescence (see Figure 7). To generate the PpIX through the 5-ALA, the concentration of 5-ALA should be d25mg/kg, and the 5-ALA should be diluted with a phosphate-buffered saline of 0.1 mL. Thus, the 5-ALA substance was given to the rat through oral intake.

Figure 7 shows the surgical environment for the animal test. As shown in Figure 7, the pen-type probe was connected to an external monitor using a cable. In the animal test of PpIX (5-ALA) fluorescence emission, the optical source of the pen-type probe was irradiated onto the tumor of a rat. The excitation wavelength (λ_ext_) and power for LD in a pen-type probe were 405 nm and 4 mW/cm^2^, respectively. The LD (Thorlabs DL-5146-101s) power was adjusted from 0 mW/cm^2^ to 19.0 mW/cm^2^, and the camera was used for SE-8J200 (endoscopy type microscopy).

The WD of a conventional surgical microscope and NIR camera (test version) is determined at least >15 cm. However, the WD of a new pen-type probe can be freely adjusted. To experiment, the WD of a pen-type probe was determined at 50 mm. Then, the pen-type probe was fixed by holding the manipulator arm for various tests.

When the LD of the pen-type probe excites a wavelength of 405 nm (4 mW/cm^2^) on the tumor, the fluorescence emitted by the PpIX in the tumor can be observed by monitoring the state for tumor removal or extant. In this experiment, the reported fluorescence emission wavelength ranged from 610 to 630 nm [13,14,17].

Figure 8 shows the results of the fluorescence emission in the tumor removal (or extant) experiments using small animals. Figure 8 shows the state of the tumor removal (or extant) through 5-ALA fluorescence in malignant tumors. The fluorescence emission images of the tumor removal (or extant) were collected using a conventional surgical microscope, an NIR camera (test version) and an infrared camera (pen-type probe). The collected images were observed on a monitor through a cable.

The results (see Figure 8) show that the pen-type probe camera can obtain an excellent imaging resolution, similar to that of an NIR camera (conventional microscope or test version NIR camera).

A fluorescent contrast agent with a half-life of 180 min (0.01 mg/kg), such as 5-ALA fluorescence, exhibits the liver-uptake phenomenon [17].

## 4. Discussion

For the observation progress of 5-ALA fluorescence emission (see Figure 8), the camera of the pen-type probe is compared with the NIR (Lumenera Lt-225 M/N). The NIR camera is a candidate for black and white color imaging, and the pen-type probe’s camera shows color imaging. The results of the pen-type probe’s camera are better than the others, as shown in Table 1.

The resolution, frame, and pixel size of a pen-type probe camera (except for the NIR camera) is excellent (more than two times) and better than that obtained in previous studies [11,13,20,21,28]. Figure 9 shows that the external irradiation optical source power (laser module) of a conventional surgical microscope was 55–160 mW/cm^2^ [4,8,11,20,28,29], and the working distance was 15–30 cm [4,8,11,16,28,29]. However, the external irradiation optical intensity will have losses of approximately 58% when the working distance is above the minimum of 200 mm [30].

To generate PpIX of 5-ALA fluorescence emission, it is necessary to increase the optical source intensity or decrease the WD. If the optical intensity of a laser module is increased, the laser module will be thermally broken because of the limitation of the laser module performance, as the laser module is weak under heat, as shown experimentally. In addition, if the working distance of a conventional surgical microscope is decreased, the field of view (FOV) dimension is narrowed. Therefore, the viewing angle of lesion observation becomes very narrow [31,32].

However, the pen-type probe LD can be sufficient for adjusting the LD power and working distance for secure lesion field of view. As shown in Figure 9, the LD and working distance of the pen-type probe were adjusted to 4 mW/cm^2^ and 50 mm, respectively, and the FOV was obtained as 60°. Thus, the tumor removal (or extant) condition is suitable for monitoring 5-ALA fluorescence emission. The conventional surgical microscope limits the adjustment range of the optical source power (55–169 mW/cm^2^) and the working distance (15–30 cm). However, the optical source and working distance of the pen-type probe are freely adjustable. In addition, the photographing angle of a conventional surgical microscope has a maximum of 30° (or 33.8°) [21,33], and the photographing angle of the pen-type probe is free (360°) [20,34,35,36,37]. Therefore, the pen-type probe does not have a blind spot, and the pen-type probe FOV angle broadens more than that of the conventional one.

The conventional surgical microscope is bulky (6 kg) [7,9,38], and the surgical microscope must be supplied with electrical energy from an outside source. Thus, it is not portable, and the size of the operating room narrows owing to the large size of the microscope. However, the irradiation of the maximum power of an LD is 19 mW, and the actual LD power is within 4 mW. The width and length of the microscopes in the study in [28] are 10.5 mm × 3.2 mm, and the size is larger more than pen-type probe. The mass of the microscope in the study in [39] is 105 g, and the width and length are 30 mm × 2.6 mm, which is heavier than the pen-type probe.

The optical source power of a proposed pen-type probe is smaller than that of the microscope in the study in [19], which is 600 mW/cm^2^. This was reduced by approximately 180 times. Thus, a pen-type probe can be operated using a battery (rechargeable). The dimension size and mass are 17 mm × 2.5 mm and 32 g. Therefore, the pen-type probe can be made small, and it is portable and can be used in a narrow operating room. The camera used for a pen-type probe is the CMOS camera. The size of a CMOS camera is smaller than the CCD camera due to the simple construction of the module. In addition, the power consumption of a CMOS camera is lower than the CCD camera, and the response time is faster than the CCD camera. The CMOS camera has a wide FOV and a dynamic range [40].

Lastly, the characteristics of the 5-ALA substance have been formally approved by the US FDA [10,11]. Therefore, it can be applied to the diagnosis of gliomas, brain cancer, breast cancer, colonic mucosa, and ovarian cancer [1,2,3,4,5].

## 5. Conclusions

In this paper, a small and portable pen-type probe was proposed. It comprises a laser diode, an endoscope camera, a brightness control device of a light source, a power supply, and a communication contactor. The pen-type probe can be connected to an external monitor using a cable. Therefore, it can be used to observe tissues from the outside by using a monitor.

If a pen-type probe is used, the shape, color, boundary of tumors, and the remanent state of the tumor can be easily distinguished through fluorescence. It allows the adjustment of the beam direction and angle; therefore, free organizational observations are possible.

The pen-type probes are independent of the working distance. Thus, there is neither an optical density loss (about the working distance) nor a blind spot (about the view angle). Laser irradiation should be conducted within a safe range of tissues. The use of lasers should be in accordance with the international medical standards (IEC 60601-2-41). Small animal experiments were conducted at the Experimental Animal Center of Korea’s Lee, Gil-Ya Cancer Diabetes Research Institute (IACUC No. LCDI-2017-0050).

For small animal experiments, the wavelength and power of the laser diode were 405 nm and 4.0 mW/cm^2^, respectively. The fluorescence emission wavelength and power were 620–670 nm and 0.4 mW/cm^2^, respectively. The fluorescence brightness duration reached a maximum of 180 min.

## Figures and Tables

**Figure 1 diagnostics-11-01014-f001:**
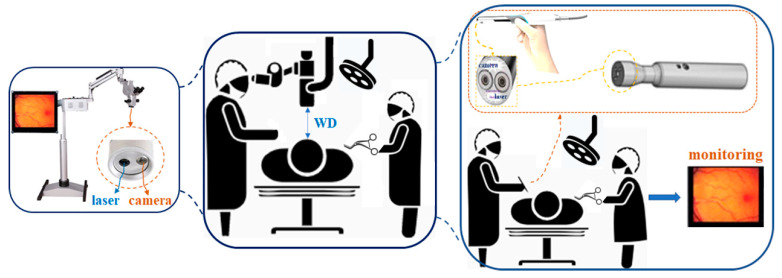
Structure of the conventional surgical microscope and the proposed pen-type probe.

**Figure 2 diagnostics-11-01014-f002:**
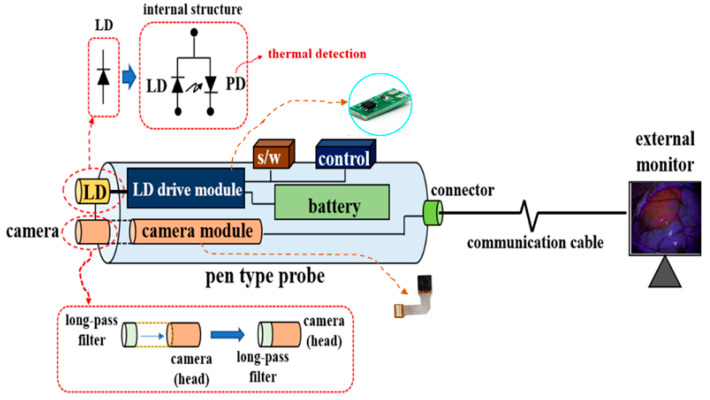
Detailed structure of a proposed pen-type probe.

**Figure 3 diagnostics-11-01014-f003:**
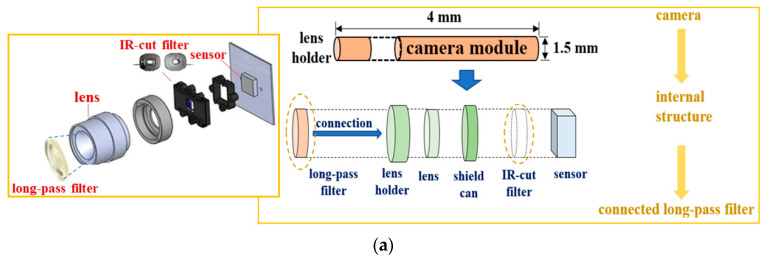
Camera module and long-pass filter (**a**) structure of the connection between the camera module and the long-pass filter (**b**) fabrication of long-pass filter.

**Figure 4 diagnostics-11-01014-f004:**
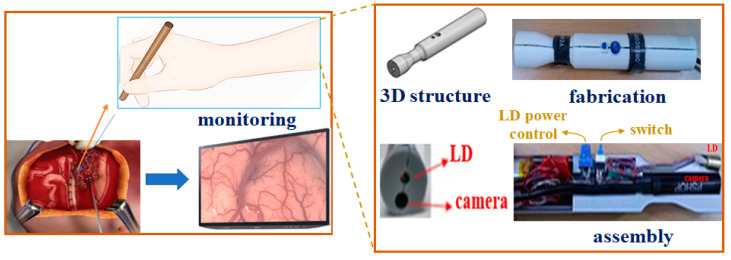
Fabricate a pen-type probe using 3D printing technique.

**Figure 5 diagnostics-11-01014-f005:**
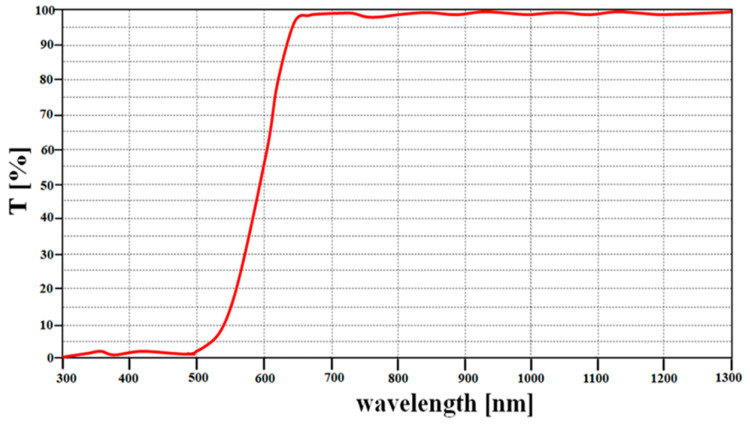
Measurement result of fabricated a long-pass filter.

**Figure 6 diagnostics-11-01014-f006:**
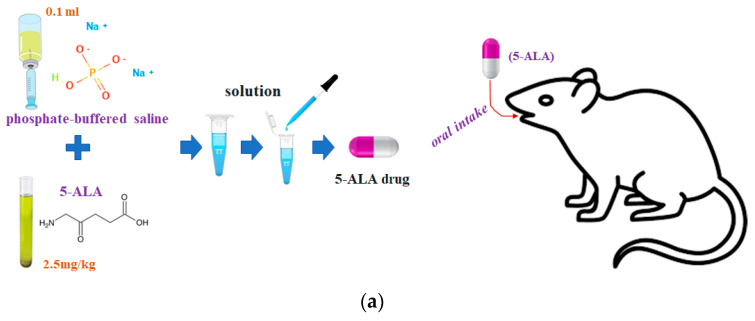
5-ALA fluorescence contrast process. (**a**) process of injection, (**b**) molecular structure of plasma protein binding by 5-ALA injection.

**Figure 7 diagnostics-11-01014-f007:**
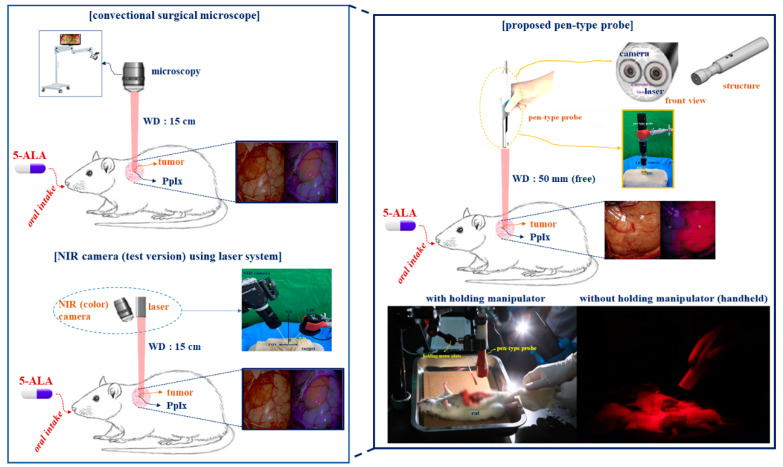
Surgery environment and comparison test for the conventional type and new pen-type probe using a small animal.

**Figure 8 diagnostics-11-01014-f008:**
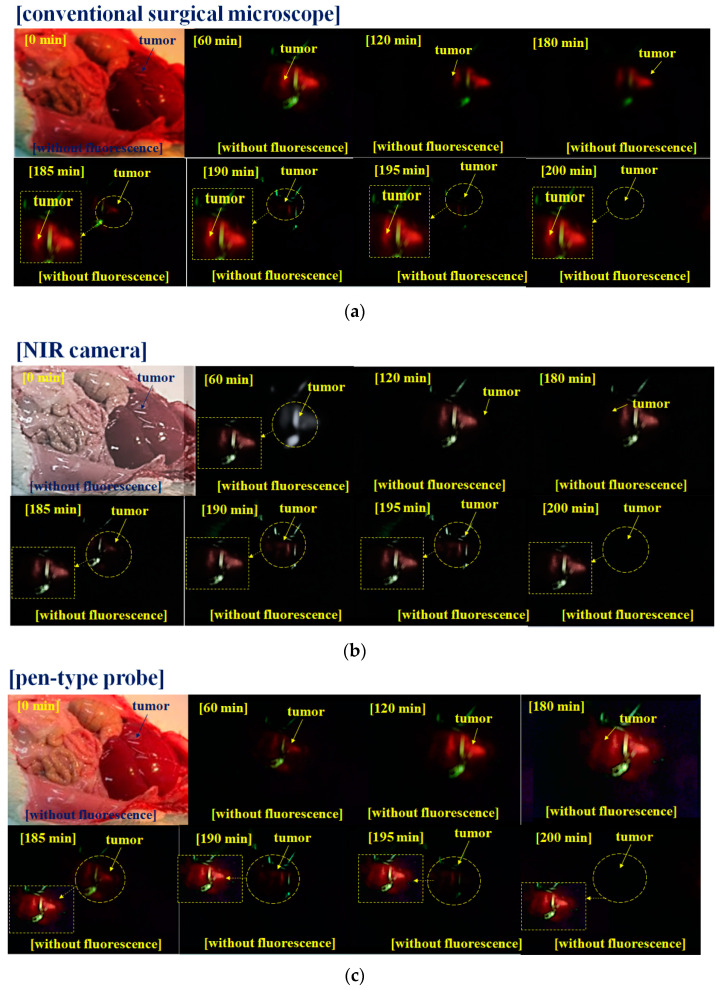
Experimental results of small animals (**a**) conventional surgical microscope (output power (excitation wavelength) and WD of 160 mW/cm^2^ (405 nm) and 15 cm) (**b**) NIR camera of Lt-225c (output power (excitation wavelength) and WD of 60 mW/cm^2^ (405 nm) and 15 cm) (**c**) pen-type probe camera (output power (excitation wavelength) and WD of 4.0 mW/cm^2^ (405 nm) and 50 mm).

**Figure 9 diagnostics-11-01014-f009:**
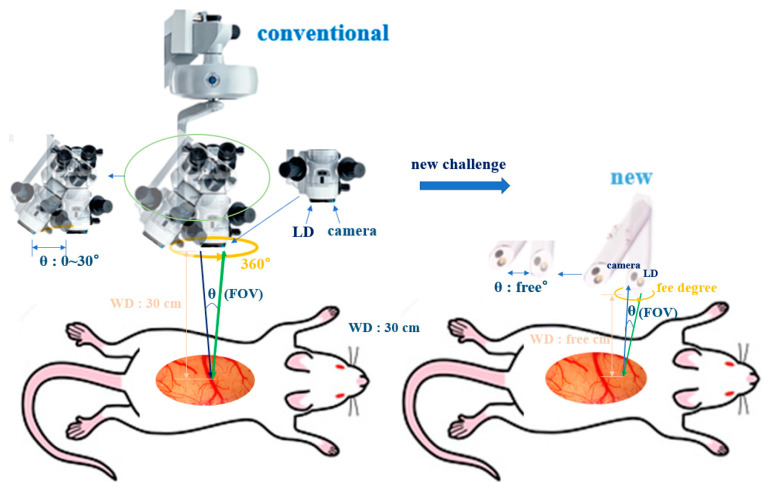
Comparison of performance for surgical microscope and a pen-type probe.

**Table 1 diagnostics-11-01014-t001:** Comparison of the performance of pen-type probe’s camera and others.

	Present Work	[20]	Fluobeam [11,21]	Spy PHI [11,21]
optical source power (mW/cm^2^)	4.0	600	382	800
WD (mm)	50.0	0.50	260	200
resolution (pixels)	1600 × 1200	640 × 480	720 × 576	252 × 512
frame rate (fps)	30	30	25	30
FOV (mm^2^)	76 × 42	1.07 × 1.07	20.0 × 14.0	19.0 × 13.0
sensor	CMOS	CCD	CCD	CCD

## Data Availability

The data presented in this study are available upon request from the permission corresponding author. The data are not publicly available because of privacy and ethical restrictions.

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
