# Peer review of "A Surgical Pen-Type Probe Design for Real-Time Optical Diagnosis of Tumor Status Using 5-Aminolevulinic Acid"

_diagnostics, 2021, doi:10.3390/diagnostics11061014_

Round 1
Reviewer 1 Report
Comments to the Editor/Authors:
- The aim of the authors’ efforts was to develop a pen-type fluorescence probe for real-time intraoperative imaging by having it consist of specifications, such as a greater working distance, that are purported to be an improvement above other standard of care imaging modalities and similar fluorescence imaging systems. The authors fabricate a fluorescence imaging device, and tout that it provides real-time imaging (just like other imaging systems) and affords pseudo-coloring of the tumor within the surgical field.
- The article is well within the scope of Diagnostics.
- The Introduction requires a major revision, as it is very unstructured and verbose while having information that is either redundant or information that should be included in a Methods/Experimental section. For example, lines 93-96 are identical to lines 77-80. Also, lines 102-106 are identical to lines 86-80. The manufacturer/commercial retailer of the filters (e.g., Thorlabs) does not need to be included in an Introduction, as this is information that should be in a Methods/Experimental section.
- The authors should not be comparing their probe against other types of imaging modalities, but they should do so against other similar fluorescence imaging modalities. There are other fluorescence imaging endoscopes and imaging systems to which the authors should be comparing their pen-type probe.
- The authors do not sufficiently identify the novelty of their work and need to do so. It should be clear how their imaging system is more advantageous than other fluorescence imaging systems and surgical microscopes. An imaging camera that has a greater FOV / working distance suffers from worse resolution (i.e., 1600 x 1200 resolution though greater FOV), which can be seen in Table 1 in the Discussion section. The information in the table should be the comparator that is used in the introduction, and not PET, CT, or MRI.
- PET similarly produces images that are overlaid with contrasting colors. PET has excellent sensitivity, but it suffers from providing poor image resolution. MRI/CT is used as the standard of care for brain cancers due to having excellent sensitivity and image resolution down to approximately 2 mm, which is similar for the latest PET instrumentation. MRI can readily distinguish anatomical structures, especially in the brain. Hence, MRI can readily distinguish tumors. PET is used for assessing metabolic activities of tumor masses.
- Figure 3. Caption. The figure is not a photograph. However, the figure is very clear and readily understandable.
- 5-ALA is delivered orally and metabolized to generate PpIX. PpIX cannot distinguish between grades I & II glioma tumor tissue from healthy tissue, if being used for glioma resection.
- Line 117. The filter that is being used as a long pass filter and not a band pass filter. Thus, the long pass filter should be only allowing for transmission of all wavelengths of light from ~650 nm and beyond (based on Figure 5) and not between 620-670 nm (which is the range of emission wavelengths that PpIX provides). The range of wavelengths that is indicated leads a reader to believe that a band pass filter is used (i.e., collects light from only 620-670 nm). 5-ALA does not provided fluorescence; its metabolite does. Also, figure 5 should not be included in the manuscript, as it is information that is obtained from the manufacturer’s / commercial retailer’s website. Also, for future reference, if an image is being utilized from elsewhere, it should be appropriately cited and permission from the publisher/journal should be obtained and stated. If the authors generated Figure 5, then there are no issues except that it should not be included in the manuscript.
- Please distinguish the difference/capabilities of a CMOS and CCD camera. Which camera type provides better imaging (resolution) and which has greater sensitivity? Why did the authors choose to utilize a CMOS? This is the type of information that should be included in a manuscript (i.e., reasoning behind their efforts).
- From an intraoperative perspective, how would it be better that the probe is a pen-type that is held in the surgeon’s hand, and thus removes the hand from being able to perform procedures? That is, the surgeon can now only have 1 free hand capable of performing excision; typically, the surgeon uses both hands for the operation. So, from a practical standpoint, how is this better?
- Figure 2 reveals a battery and it is discussed in the text. Is the pen-type probe rechargeable or require new batteries when the battery expires? Typically, most endoscopes have the image communication line incorporate the power line in an all-in-one bundle.
- During procedures, “white” fluorescent light is emitted from a light source and provides reflection in the surgical cavity. I see no reflection of a natural light source that is used during the authors’ surgeries that they performed. It appears that the surgery was performed under no light, which does not serve as a good comparison to how such instruments are typically used. Was the procedure performed in the dark aside from having the LD on?
Author Response
Comments and Suggestions for Authors
Comments to the Editor/Authors:
1. The aim of the authors’ efforts was to develop a pen-type fluorescence probe for real-time intraoperative imaging by having it consist of specifications, such as a greater working distance, that are purported to be an improvement above other standard of care imaging modalities and similar fluorescence imaging systems. The authors fabricate a fluorescence imaging device, and tout that it provides real-time imaging (just like other imaging systems) and affords pseudo-coloring of the tumor within the surgical field.
2. The article is well within the scope of Diagnostics.
I am glad to incorporate requirement of modification in the manuscript.
Thank you for your comments.
3. The Introduction requires a major revision, as it is very unstructured and verbose while having information that is either redundant or information that should be included in a Methods/Experimental section. For example, lines 93-96 are identical to lines 77-80. Also, lines 102-106 are identical to lines 86-80. The manufacturer/commercial retailer of the filters (e.g., Thorlabs) does not need to be included in an Introduction, as this is information that should be in a Methods/Experimental section.
I decided to delete the lines 77 to 90 and production names (ex. DL-5146-101s and SE8J200 (lines 69)). I revised the lines 81 to 91.
The excitation wavelength of a LD is 405 nm, and the bias voltage / current, output power, and beam divergence (θdiv) are 5.6 V / 100 mA, 19.0 mW/cm2, and 16°, respectively. The laser brightness control performance was controlled using a variable resistor (max. 10 kΩ). Therefore, the LD can adjust the excitation power. The endoscope camera was used in the CMOS sensor, and the bias voltage of the camera module was 3 V. In addition, the horizontal length and diameter of the camera dimensions were 4 mm and 1.5 mm, respectively, as shown in Fig. 3 (a).
From the figure, the resolution, imaging frame, pixel size, field of view (FOV), and sensor type of the camera were 1600×1200, 30 fps, 2.0 Mpixel, 60°, and CMOS sensor type, respectively. The irradiation beam angle of the LD was free degree. In addition, the WD and beam focus angle (θf) of the camera were 50 mm and 9.5°, respectively (see Fig. 1). Then, the WD was adjusted from 0 to 50 cm, and the realization environment was measured to be 50 mm.
4. The authors should not be comparing their probe against other types of imaging modalities, but they should do so against other similar fluorescence imaging modalities. There are other fluorescence imaging endoscopes and imaging systems to which the authors should be comparing their pen-type probe.
I added the lines 52 to 63, Table 1, and lines 218 to 223 about the handheld type surgical microscope. Please Refer to red words of lines number and Table 1.
# word lines 52 to 63
To overcome these problems, several literatures of handheld surgical microscope still have been reported [19]-[27].
The advantage of handheld type surgical microscope is smaller than conventional surgical microscope, and the WD is shorter than conventional one. In addition, the handheld type surgical microscope is freely of photographing angle. However, the handheld type surgical microscope is still bulk [19], [25, 26]. Especially, the weight of [25] is light. However, the size is long. Furthermore, the WD of [20] using image-guided raman spectroscopy technique is shortness. But, the power of optical source at excitation wavelength for [20] is still used for highly energy (>600 mW @ 405 nm). Also, the system is not integrated with optical source and camera. They are independent of each other. Moreover, the construction of a system is complex. Therefore, the system is inconvenient for the surgeon to use the camera and light source, respectively. It also takes a lot of hands for the operation.
# Refer to red words of Table 1.
# word lines 218 to 223
The size of [28] is 10.5 mm x 3.2 mm. However, the weight is 105 g it is very heavy. In additional, the weight of [29] is 0.3 g which is very light. However, the dimension size is bulky which is 30.0 mm x 2.6 mm.
The optical source power of a proposed pen-type probe is very smaller than [19]. The [19] power is 600 mW/cm2.
5. The authors do not sufficiently identify the novelty of their work and need to do so. It should be clear how their imaging system is more advantageous than other fluorescence imaging systems and surgical microscopes. An imaging camera that has a greater FOV / working distance suffers from worse resolution (i.e., 1600 x 1200 resolution though greater FOV), which can be seen in Table 1 in the Discussion section. The information in the table should be the comparator that is used in the introduction, and not PET, CT, or MRI.
Thank you for your comments. I deleted about PET, CT, and MRI, and I added the handheld types surgical microscope instead of PET, CT, and MRI. Please refer to word lines 39-63 (above all 52 to 63).
# word lines 39-63
Therefore, a fluorescence observation module can be connected to a surgical color-imaging camera. When used in surgery, fluorescent image guidance cameras for surgery can distinguish the shape, color, and boundaries of tumors and blood vessels because of their fluorescence expression [6, 7]. However, these guidance cameras have disadvantages as they are large and heavy (camera head: above 4 kg, entire: above 6 kg) with limited portability [8]-[14]. In addition, they allow limited adjustment of the observed beam angles (maximum 30°) and working distance (WD) of 15–30 cm/cm2 [8]-[17]. Because a surgical camera uses a high-energy light source (55–160 mW/cm2) for the electromotive force of fluorescence emission, optical source beam energy loss (above 58%) can occur at the WD (15–30 cm), and strong ambient light can be observed around the tissue (photobleaching) [8], [16]-[18]. Thus, the surgical microscope is difficult to operate for tissue observation owing of the limitations of beam WD control and adjustment of the left and right of the camera [9, 10]. In addition, surgical diagnostic systems are not portable because of their large size [16], [18].
To overcome these problems, Several literatures of handheld surgical microscope still have been reported [19]-[27].
The advantage of handheld type surgical microscope is smaller than conventional surgical microscope, and the WD is shorter than conventional one. In addition, the handheld type surgical microscope is freely of photographing angle. However, the handheld type surgical microscope is still bulk [19], [25, 26]. Especially, the weight of [25] is light. However, the size is long. Furthermore, the WD of [20] using image-guided raman spectroscopy technique is shortness. But, the power of optical source at excitation wavelength for [20] is still used for highly energy (>600 mW @ 405 nm). Also, the system is not integrated with optical source and camera. They are independent of each other. Moreover, the construction of a system is complex. Therefore, the system is inconvenient for the surgeon to use the camera and light source, respectively. It also takes a lot of hands for the operation.
6. PET similarly produces images that are overlaid with contrasting colors. PET has excellent sensitivity, but it suffers from providing poor image resolution. MRI/CT is used as the standard of care for brain cancers due to having excellent sensitivity and image resolution down to approximately 2 mm, which is similar for the latest PET instrumentation. MRI can readily distinguish anatomical structures, especially in the brain. Hence, MRI can readily distinguish tumors. PET is used for assessing metabolic activities of tumor masses.
Thank you for your good advice. Your advice was excellent, and I am glad I took it. I deleted about PET, CT, and MRI, and I added the handheld types surgical microscope instead of PET, CT, and MRI. Please refer to word lines 39-63 (above all 52 to 63).
7. Figure 3. Caption. The figure is not a photograph. However, the figure is very clear and readily understandable.
That’s right. I am disposed to agree with you. So. I changed Figure caption (title) and photograph about fabricated filter. Please refer to Figure 3.
8. 5-ALA is delivered orally and metabolized to generate PpIX. PpIX cannot distinguish between grades I & II glioma tumor tissue from healthy tissue, if being used for glioma resection.
Thank you for your good advice. Your advice was excellent, and I am glad I took it. I revised the PpIx. Please refer to word lines 131-137 and Fig. 6 and 7. I revised entire pages about PpIx.
# word lines 131-137
The 5-ALA substance is generated Protoporphyrin IX (PpIx) material after 5 to 6 hours through the chemical reaction with tumor when take medication (oral intake) as shown in Fig. 5 (a) and (b) [17]. At this time, the PpIx has not response of fluorescence emission property. However, If the external light source is irradiated in the PpIx, the PpIx is emitted the fluorescence (see Fig. (b) and (c)). To generate the PpIx through the 5-ALA, the concentration of 5-ALA is determined 25mg/kg, and the 5-ALA is diluted with phosphate-buffered saline of 0.1 ml. Thus, the 5-ALA substance was oral intake of rat.
9. Line 117. The filter that is being used as a long pass filter and not a band pass filter. Thus, the long pass filter should be only allowing for transmission of all wavelengths of light from ~650 nm and beyond (based on Figure 5) and not between 620-670 nm (which is the range of emission wavelengths that PpIX provides). The range of wavelengths that is indicated leads a reader to believe that a band pass filter is used (i.e., collects light from only 620-670 nm). 5-ALA does not provided fluorescence; its metabolite does. Also, figure 5 should not be included in the manuscript, as it is information that is obtained from the manufacturer’s / commercial retailer’s website. Also, for future reference, if an image is being utilized from elsewhere, it should be appropriately cited and permission from the publisher/journal should be obtained and stated. If the authors generated Figure 5, then there are no issues except that it should not be included in the manuscript.
I revised the word lines 117. Thank you for your advice. Please refer to word lines 112-114 and 102-104. I revised the PpIx (see 127 to 137 or whole pages).
In the Figure 5, I was designed the long-pass filter. Because the long-pass filter is not sold the high performance of small size for pen-type probe applications. Thus, the filter is fabricated on glass coated substrate using SiO2 and Ti3O5. I was measured to our laboratory using by spectrometer.
# word lines 112-114
The measurement results for the fabrication of the long-pass filter are shown in Fig. 5. From the figure, the cut-in wavelength, transmission (T), and reflection (OD: Optical density) are 600 nm, 98%, and 0.2%, respectively.
# word lines 102-104
The long-pass filter only transmits above 600 nm which transmission wavelength range above 600 nm sufficiently transmit the PpIx (5-ALA) emission wavelength range. Because the PpIx emission wavelength range is 620–670 nm through the excitation wavelength of 405 nm.
10. Please distinguish the difference/capabilities of a CMOS and CCD camera. Which camera type provides better imaging (resolution) and which has greater sensitivity? Why did the authors choose to utilize a CMOS? This is the type of information that should be included in a manuscript (i.e., reasoning behind their efforts).
CMOS camera has good performance more than CCD camera which is suitable for use in surgery. Because the CMOS camera has been small size of device module, simple construction of circuit, rapidly response time, low power consume, and wide FOV / dynamic range. Please refer to word lines 225-229.
# word lines 225-229
Therefore, the pen-type probe can be made small, and it is portable and can be used in a narrow operating room. The camera of a pen-type probe is used for CMOS camera. The size of a CMOS camera is smaller than the CCD camera due to simple construction of module. In addition, the power consume of a CMOS camera is lower more than CCD camera, and the response time is very faster better than CCD camera. The CMOS camera has wide FOV and dynamic range [42].
11. From an intraoperative perspective, how would it be better that the probe is a pen-type that is held in the surgeon’s hand, and thus removes the hand from being able to perform procedures? That is, the surgeon can now only have 1 free hand capable of performing excision; typically, the surgeon uses both hands for the operation. So, from a practical standpoint, how is this better?
Yes. It’s correct. The surgeon clasp both hands the surgical instrument while the operation time. But, a pen-type probe is applied to very short time for observation of tumor removal or extant status of a tumor. To observe the tumor status, the conventional surgical microscope is heavy which experience inconvenience during the short observation time. In addition, the conventional surgical microscope is narrowed the field of view and it is not easy for adjustment of working distance. Therefore, the handheld surgical microscope is convenient for Instantaneous observation.
Figure 2 reveals a battery, and it is discussed in the text. Is the pen-type probe rechargeable or require new batteries when the battery expires? Typically, most endoscopes have the image communication line incorporate the power line in an all-in-one bundle.
Yes. I used the rechargeable battery. The ‘wire’ is cable which cable is connected camera to outside monitor in the Figure 6.
The future plan is challenge of wireless communication between monitor and camera.
# word lines 223-224
This was reduced by approximately 180 times. Thus, a pen-type probe can be operated using a battery (rechargeable).
12. During procedures, “white” fluorescent light is emitted from a light source and provides reflection in the surgical cavity. I see no reflection of a natural light source that is used during the authors’ surgeries that they performed. It appears that the surgery was performed under no light, which does not serve as a good comparison to how such instruments are typically used. Was the procedure performed in the dark aside from having the LD on?
Filter is excellent of performance. Thus, the filter can not transmit excitation wavelength and others, and the filter is transmitted only fluorescence emission wavelength. So, we can try surgery and observation of fluorescence emission status when the switch turns ON status surgery room’s lights. However, we were turn off the light switch. Because we was observed with fluorescence dye and PpIx metabolic emission of 5-ALA status of tumor on the Rat surgery affected area by naked eye during the fluorescence emission test. The fluorescence dye status can be observed through the naked eye with dark states. So, We was obliged to shooting in the dark states.
Thank you very much for your good advice. Because we will be observation for fluorescence emission test in the bright place on next plan.
Figure 6 is not picture for comparison. In the picture, right side of picture is experiment with handheld type. The left side picture is used for holding manipulator because we try to various test (see word lines 158-161). White light is smart phone’s light. Because we cannot operate the performance control (push of buttons) for monitoring capture due to dark view. The observation for fluorescence emission was tested in the dark environment (light off).
# word lines 158-161
The WD of a conventional surgical microscope and NIR camera is determined at least >15 cm. However, the WD of a pent-type probe is adjust freely available. To experiment, the WD of a pen-type probe is determined 50 mm. Then, the pen-type probe is fixed by holding manipulator arm for various test.
Submission Date
20 April 2021
Date of this review
07 May 2021 19:01:53

Reviewer 2 Report
Overview:
In this article, the authors propose a compact pen-type probe with a portable surgical fluorescence microscope. It comprises a laser diode, an endoscope camera, a brightness control device of a light source, a power supply, and a communication contactor. This study develop a pen-type surgical fluorescence microscope that is small and portable with an adjustable beam angle and provides real-time diagnosis. The performance of the proposed pen-type probe was verified through animal experiments.
Although this paper does address a relevant and timely issue and is designed logically, is well written and the details for the experiments are sufficient. However, there's several minor changes could be addressed before this article could be acceptable for publication.
Specific comments:
- Although the tumor removal (or extant) condition is suitable for monitoring the fluorescence emission, the author may should give the absorption and emission spectra of 5-ALA and PpIx to facilitate readers to understand more clearly. The detected fluorescence is from PpIX, not 5-ALA.
- Page 9, line 202: Figure 10 appears, but there is no Figure 10 in the article.
- Figure 8: Scale bar should be added. And the images of tumor seem not in the same scale as others. What does the green color represent?
- There is also a need for copy editing for the entire manuscript to correct grammatical errors and restructure sentences appropriately.
Author Response
Specific comments:
- Although the tumor removal (or extant) condition is suitable for monitoring the fluorescence emission, the author may should give the absorption and emission spectra of 5-ALA and PpIx to facilitate readers to understand more clearly. The detected fluorescence is from PpIX, not 5-ALA.
à I am glad to incorporate requirement of modification in the manuscript.
à Thank you for your comments.
à Thank you for your good advice. Your advice was excellent, and I am glad I took it. I revised the PpIx. Please refer to word lines 131-137 and Fig. 6 and 7. I revised entire pages about PpIx.
à I added the measurement results for spectrum range of absorption and emission that is expressed the 405 nm (excitation wavelength) and 610 nm (peak wavelength / PpIx for 5-ALA fluorescence). Please refer to Fig. 7 (b).
# word lines 131-137
The 5-ALA substance is generated Protoporphyrin IX (PpIx) material after 5 to 6 hours through the chemical reaction with tumor when take medication (oral intake) as shown in Fig. 5 (a) and (b) [17]. At this time, the PpIx has not response of fluorescence emission property. However, If the external light source is irradiated in the PpIx, the PpIx is emitted the fluorescence (see Fig. (b) and (c)). To generate the PpIx through the 5-ALA, the concentration of 5-ALA is determined 25mg/kg, and the 5-ALA is diluted with phosphate-buffered saline of 0.1 ml. Thus, the 5-ALA substance was oral intake of rat.
- Page 9, line 202: Figure 10 appears, but there is no Figure 10 in the article.
à Thank you for your comments. I revised the Figure 9. Figure 10 is deleted. Please refer to 190-217 and Figure 9.
# word lines 190-217
In Fig. 9, the external irradiation optical source power (laser module) of a conventional surgical microscope was 55–160 mW/cm2 [4], [8], [11], [20], [28], [32], and the working distance was 15–30 cm [4], [8], [11], [16], [28], [32]. However, the external irradiation optical intensity will have losses of approximately 58 % when the working distance is above the minimum of 200 mm [33].
To generate PpIx of 5-ALA fluorescence emission, it is necessary to increase the optical source intensity or decrease the WD. If the optical intensity of a laser module is increased, the laser module will be thermally broken because of the limitation of the laser module performance as the laser module is weak to heat, as shown experimentally. In addition, if the working distance of a conventional surgical microscope is decreased, the field of view (FOV) dimension is narrowed. Therefore, the viewing angle of lesion observation becomes very narrow [34, 35].
However, the pen-type probe LD can be sufficient for adjusting the LD power and working distance for secure lesion field of view. As shown in Fig. 9, the LD and working distance of the pen-type probe are adjusted to 4 mW/cm2 and 50 mm, respectively, and the FOV can be obtained as 60°. Thus, the tumor removal (or extant) condition is suitable for monitoring 5-ALA fluorescence emission. The conventional surgical microscope limits the adjustment range of the optical source power (55–169 mW/cm2) and the working distance (15–30 cm). However, the optical source and working distance of the pen-type probe are freely adjustable. In addition, the photographing angle of a conventional surgical microscope has a maximum of 30° (or 33.8°) [21, 36], and the photographing angle of the pen-type probe is free (360°) [20], [37]-[40]. Therefore, the pen-type probe does not have a blind spot, and the pen-type probe FOV angle broadens more than that of the conventional one.
The conventional surgical microscope is bulky (6 kg) [7], [9], [41], and the surgical microscope must be supplied with electrical energy from an outside source. Thus, it is not portable, and the size of the operating room narrows owing to the large size of the microscope. However, the irradiation of the maximum power of an LD is 19 mW, and the actual LD power is within 4 mW.
- Figure 8: Scale bar should be added. And the images of tumor seem not in the same scale as others. What does the green color represent?
à I added scale picture. Please refer to Fig. 8. Half-life time of fluorescence emission is 180 minutes. Then, the fluorescence emission status has dwindled away to nothing after 180 minutes. Thus, the tumor size has any difference due to darkness.
à green color is not PpIx (5-ALA). It is fluorescence sodium. We was tested fluorescence emission phenomenon using fluorescence sodium (yellow dye) before PpIx (5-ALA) fluorescence emission test. The green color and PpIx fluorescence emission imaging are not any relation.
- There is also a need for copy editing for the entire manuscript to correct grammatical errors and restructure sentences appropriately.
à Thank you for your comments. I am currently requesting correction of English grammar through English correction expert.
Submission Date
20 April 2021
Date of this review
19 May 2021 09:55:13

Round 2
Reviewer 1 Report
Comments to the Editor/Authors (round 2):
- The authors have appropriately addressed nearly all the comments that were suggested in the previous round of review. However, many of their responses should be placed into the text, not just into the cover letter.
- This manuscript and the information therein have the potential to be very good and informative, if the authors can organize their introduction a bit better and clean up the English throughout the manuscript. I suggest that a 3rd party whose native language is English review the paper.
- The grammar throughout the manuscript does require significant editing to be suitable for publication. Below I mention a few examples along with additional comments.
- Line 39 can connect directly with Line 40 (i.e., there should not be 2 paragraphs, but only 1).
- Lines 39-42 need to be rewritten to properly express the idea that they are trying to relay.
- It appears that Lines 42-45 are similar in nature to Lines 48-51. There is redundancy in the information that is provided.
- Lines 56-60 need to be cleaned up to make better sense.
- As the thesis statement, Lines 64-67 should be clearer.
- The 2nd part of line 84 needs to be rephrased.
- Line 89. What is “free degree”? Could you please address or rewrite so that it makes sense?
- Line 98. Shouldn’t it be “cut-in”? Please make the correction if so.
- Lines 102-104 need to be rewritten to make sense. I believe the statement in Lines 103-104 is not a sentence.
- The 2 panels on the left in Figure 4 should be moved to the introduction, whereby the 2 panels on the right in Figure 4 should now become Figure 4.
- In line 110, please insert “significantly” before “lighter”.
- Figure 6(a). For the conventional surgical microscope and the conventional test set using the NIR and laser system, are the raw images supposed to be in opposite locations? That is, shouldn’t the conventional test set using the NIR and laser system produce the fluorescence image and the conventional surgical microscope produce the black/white only images?
- What information are the authors trying to relay in Figure 6, as it’s not clear to me?
- Beginning at Line 131, “PpIx” should be “PpIX”. The “X” is capitalized. Please make the corrections throughout the manuscript.
- Figure 7a, right panel. An absorption spectrum should be shown with the excitation wavelength at 405 nm. An emission spectrum should be shown with the emission of PpIX when irradiated at 405 nm.
- Line 148. Shouldn’t “lumenera” be capitalized to be “Lumenera”?
- Lines 146-153 are confusing. Can the authors simply rewrite the paragraph?
- Lines 158-161 describe Figure 6(b). The material should be placed near the graphic.
- How is Figure 7(c) different than Figure 6(a) or Figure 6(b). That is, what new information is the graphic providing?
- Line 159. “pent-type” should be “pen-type”.
- Lines 167-168 should be combined into 1 sentence, as they both express the same idea.
- The WD and optical source power used by each system should be included in Table 1.
- Line 218 is not a complete sentence. Could the authors please correct?
- Line 222 is not a complete sentence. Could the authors please correct?
- Line 224. “weigh” should be “mass”.
Author Response
Point 1: The authors have appropriately addressed nearly all the comments that were suggested in the previous round of review. However, many of their responses should be placed into the text, not just into the cover letter.
Response 1: I am glad to incorporate requirement of modification in the manuscript. I shall be pretty well reflecting to your comments.
Point 2: This manuscript and the information therein have the potential to be very good and informative, if the authors can organize their introduction a bit better and clean up the English throughout the manuscript. I suggest that a 3rd party whose native language is English review the paper.
Response 2: I received already revision for English grammar through the Editage. Notwithstanding, I am currently requesting correction of English grammar through English correction expert. I will make sure to revise the English as a whole before publishing. And I shall be pretty well reflecting to your comments.
Point 3: The grammar throughout the manuscript does require significant editing to be suitable for publication. Below I mention a few examples along with additional comments.
Response 3 : I received already revision for English grammar through the Editage. Notwithstanding, I am currently requesting correction of English grammar through English correction expert. I will make sure to revise the English as a whole before publishing. And I shall be pretty well reflecting to your comments.
Point 4: Line 39 can connect directly with Line 40 (i.e., there should not be 2 paragraphs, but only 1).
Response 4: Thank you for your good advice. I revised about your comments. Please refer to manuscript of word lines 38.
Point 5: Lines 39-42 need to be rewritten to properly express the idea that they are trying to relay.
Response 5: Thank you for your good advice. I revised about your comments. Please refer to manuscript of word lines 38-41. This sentence will be corrected in English by an expert. Please refer to it. Currently, I have requested an English proofreading application to an English proofreading expert.
Point 6: It appears that Lines 42-45 are similar in nature to Lines 48-51. There is redundancy in the information that is provided.
Response 6: Thank you for your good advice. I deleted the word lines of 48-51. Because the word lines of 48-51 seems to overlap with 41-44.
Point 7: Lines 56-60 need to be cleaned up to make better sense.
Response 7: I revised the word lines 56-60. Thank you for your advice. Please refer to word lines 52-57.
Point 8: As the thesis statement, Lines 64-67 should be clearer.
Response 8: I revised the lines 64-67. Please refer to word lines of 60-68. This sentence will be corrected in English by an expert. Please refer to it. Currently, I have requested an English proofreading application to an English proofreading expert.
Point 9: The 2nd part of line 84 needs to be rephrased.
Response 9: I revised about 84 lines. Please refer to word lines of 84-85.
Point 10: Line 89. What is “free degree”? Could you please address or rewrite so that it makes sense?
Response 10: I revised the 89 lines as following:
“The irradiation beam angle of a LD can be adjusted by 360°.”
Please refer to word lines of 89. Thank you.
Point 11: Line 98. Shouldn’t it be “cut-in”? Please make the correction if so.
Response 11: Cut-on is expressed a lot in textbooks and papers. However, cut-in is also expressed a lot. But I will use cut-in. I advised the cut-in instead of cut-on. Thank you very much. Please refer to word line of 113.
Point 12: Lines 102-104 need to be rewritten to make sense. I believe the statement in Lines 103-104 is not a sentence.
Response 12: I revised the English grammar. Please refer to word lines of 102-105.
Point 13: The 2 panels on the left in Figure 4 should be moved to the introduction, whereby the 2 panels on the right in Figure 4 should now become Figure 4.
Response 13: I decided that I would not need the picture. The reason is that the contents of the surgical microscope already exist in the introduction. Therefore, the picture is not needed in the handheld type picture.
Point 14: In line 110, please insert “significantly” before “lighter”.
Response 14: I inserted the significantly before lighter. Please refer to word lines of 110.
Point 15: Figure 6(a). For the conventional surgical microscope and the conventional test set using the NIR and laser system, are the raw images supposed to be in opposite locations? That is, shouldn’t the conventional test set using the NIR and laser system produce the fluorescence image and the conventional surgical microscope produce the black/white only images?
Response 15: This picture seems to have been mistaken. Therefore, I changed the picture again to be correct. And the NIR camera is also a color version, not black and white. Thank you very much for the point. Please refer to Figure 7.
I changed the position of Fig. 6 and Fig. 7.
Point 16: What information are the authors trying to relay in Figure 6, as it’s not clear to me?
Response 16: The performance of the NIR camera of the existing surgical microscope and the test version is similar. However, a comparison picture shows that the performance and image quality of the pen-type probe are better than conventional one.
Point 17: Beginning at Line 131, “PpIx” should be “PpIX”. The “X” is capitalized. Please make the corrections throughout the manuscript.
Response 17: Thanks for the deep advice. I changed all to PpIX.
Point 18: Figure 7a, right panel. An absorption spectrum should be shown with the excitation wavelength at 405 nm. An emission spectrum should be shown with the emission of PpIX when irradiated at 405 nm.
Response 18: Thank you for your comments. I revised the figure. I re-measured and I changed the result. Please refer to Figure 6 (b).
I changed the position of Fig. 6 and Fig. 7.
Point 19: Line 148. Shouldn’t “lumenera” be capitalized to be “Lumenera”?
Response 19: Thank you for your comment. I changed Lumenera. Thanks again. Please refer to word line of 178.
Point 20: Lines 146-153 are confusing. Can the authors simply rewrite the paragraph?
Response 20: The texts were changed entirely. And some of the texts were deleted and organized. In addition, some of the content has been moved to the title of Figure 8 (a) and (b).
Please refer to word lines of 136-141 and Figure 8 (a) and (b).
Point 21: Lines 158-161 describe Figure 6(b). The material should be placed near the graphic.
Response 21: I revised the word lines of 158-161 near Fig. 6. Please refer to word lines of 142-145.
Point 22: How is Figure 7(c) different than Figure 6(a) or Figure 6(b). That is, what new information is the graphic providing?
Response 22: Figure 7 (c) and Figure 6 (b) seem to overlap. So I deleted Figure 7(c). So, this is the content to compare with Figure 6 (a) (or b) and Figure 6 (c) (same as Figure 7 (c)) In other words, it is a picture to help understand the difference between the conventional system and the new system (conventional and handheld). Figure 6 and Figure 7 are summarized. Please refer to Figure 6 and 7.
Point 23: Line 159. “pent-type” should be “pen-type”.
Response 23: I revised the pen-type. Please refer to word lines of 143.
Point 24: Lines 167-168 should be combined into 1 sentence, as they both express the same idea.
Response 24: I revised the one sentence. Please refer to world lines of 161-162.
Point 25: The WD and optical source power used by each system should be included in Table 1.
Response 25: I added the optical source and WD in the Table 1. Please refer to Table 1. Thank you for your advice.
Point 26: Line 218 is not a complete sentence. Could the authors please correct?
Response 26: I revised into the correctly sentence. Please refer to 218-221. This sentence will be corrected in English by an expert. Please refer to it. Currently, I have requested an English proofreading application to an English proofreading expert.
Point 27: Line 222 is not a complete sentence. Could the authors please correct?
Response 27: I revised into the correctly sentence. Please refer to word lines of 216-217.
Point 28 : Line 224. “weigh” should be “mass”.
Response 28 : I revised the mass instead of weight. Please refer to word lines of 218. Thank you for your advice.
Currently, I have requested an English proofreading application to an English proofreading expert. Thank you for your good comments.
